# REINFORCED ADAPTIVE ROUTING FOR MIXTURE-OF-EXPERT MODELS

## ABSTRACT

With the rapid development of large language models (LLMs), the mixture-of-experts (MoE) architecture attracts increasing attention due to its advantages in scaling capacity and enhancing performance. However, MoE requires activating multiple experts during training and inference, which introduces substantial computational and memory overhead. This makes acceleration essential in resource-constrained or latency-sensitive settings. Existing adaptive expert selection approaches often rely on heuristics or single-source supervision, lacking a unified formulation that simultaneously captures accuracy, balanced utilization, and efficiency. To address this, we propose a reinforcement learning–based adaptive routing approach that integrates a policy network into the standard MoE framework and optimizes expert selection end-to-end with a multi-objective reward. Experiments on benchmark datasets demonstrate that our approach substantially improves training efficiency while maintaining accuracy and promoting more balanced expert utilization.[1]

## 1 INTRODUCTION

As large language models (LLMs) evolve, researchers continually devise architectural enhancements to elevate model capabilities (Naveed et al., 2023; Webb et al., 2023; Egressy & Stühmer, 2025). Among these, the mixture-of-experts (MoE) architecture (Shazeer et al., 2017), which performs routing computations across specialized experts, stands out for expanding LLMs' capacity and delivering strong performance in a wide range of tasks (Riquelme et al., 2021; Fedus et al., 2022; Lin et al., 2024; Do et al., 2025; Zhou et al., 2025). However, during both training and inference, MoE still activates numerous experts per token, resulting in substantial computational and memory overhead (Rajbhandari et al., 2022). In resource-constrained or latency-sensitive environments, this bottleneck critically limits the practicality and deployment viability of MoE. Therefore, it is essential to optimize the computational cost of MoE models while preserving model performance (Zhou et al., 2022; Huang et al., 2023; Liu et al., 2024; He et al., 2024; Sun et al., 2025).

Existing studies generally perform adaptive expert selection guided by task context or token difficulty to dynamically allocate computational resources for efficient training and inference of MoE-based LLMs. Most approaches typically follow two main directions: width-based and depth-based adaptation (Lepikhin et al., 2020; Rajbhandari et al., 2022; Raposo et al., 2024; Huang et al., 2024b; Aghdam et al., 2024; Jin et al., 2024). In the width dimension, efforts focus on enabling the router to dynamically determine the number of experts to activate for each token (Huang et al., 2024b; Aghdam et al., 2024; Jin et al., 2024). Conversely, depth-based approaches save computation by skipping certain Transformer layers and directly forwarding inputs to deeper layers (Lepikhin et al., 2020; Rajbhandari et al., 2022; Raposo et al., 2024). In practice, many approaches integrate both strategies. For example, the mixture-of-depths (MoD) framework (Raposo et al., 2024) selects which experts to activate and which layers to skip. Other studies introduce specialized "null experts" as placeholders in the original MoE pool; selecting these experts effectively skips computation or reduces the number of active experts (Zeng et al., 2024; Jin et al., 2024). However, most existing approaches rely solely on downstream task performance as their supervisory signal to train the expert selection mechanism. It is hard for the single-source supervision to finely control the number of experts activated per layer, potentially increasing computational cost without delivering notable

---

[1]Code will be released in the final version of the paper.

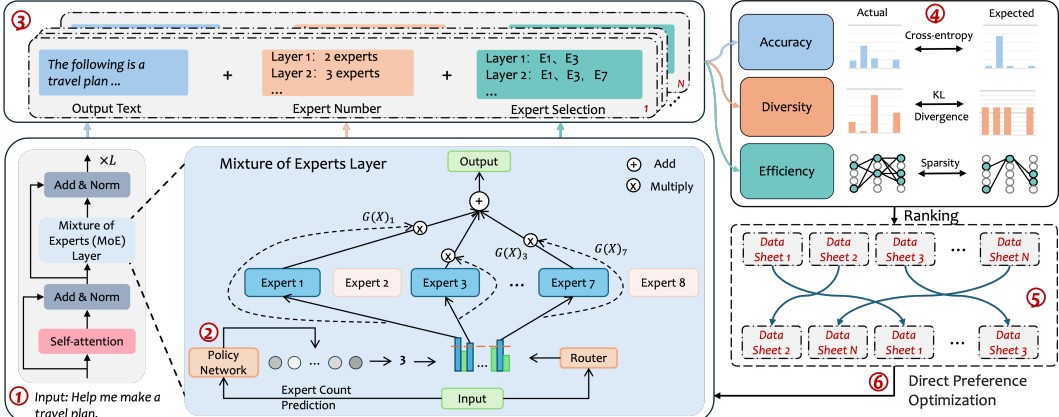

Figure 1: The overall architecture of our approach. The bottom-left presents the MoE module with the policy networks to predict the number of activated experts. The top-left presents the output text, the predicted expert number at different layers for various tokens, and the exact activated experts that process the tokens. The top-right presents the process to compute the rewards for assessing the quality of the output text, the number of active experts, and the selected experts. The outputs are ranked based on the reward to construct the preference data to optimize the LLM with DPO.

performance gains. Therefore, it is necessary to introduce additional constraint variables to guide expert selection, allowing the model to allocate resources more efficiently while maintaining accuracy (Muzio et al., 2024; Qiu et al., 2025). For example, an intuitive supervisory signal is to use a predefined expert activation ratio to constrain the actual number of activated experts. Additionally, it is generally expected to have a uniform activation distribution across experts to prevent a few experts from shouldering excessive computation. As these auxiliary signals are challenging to integrate through conventional supervised learning, we consider employing reinforcement learning (RL) to incorporate and optimize these constraints.

In this paper, we propose an RL-based dynamic routing approach to enhance training and inference efficiency in MoE models. Specifically, we integrate a policy network into the standard MoE framework to predict the number of experts to activate at each layer. Based on the policy network's prediction, we select the top-scoring experts to process each token. We then feed the activated experts' identities and counts, along with the model's final output, into a reward function for evaluation. The reward function scores from three perspectives: output accuracy, diversity of activation distribution, and alignment of activation count with the predefined target. We generate preference data by ranking different outcomes based on the rewards. We train the router and the policy network using a direct preference optimization (DPO) (Rafailov et al., 2023) strategy. We evaluate our approach on a wide range of benchmark datasets, and the results and analyses show that it reduces computation and improves runtime efficiency without sacrificing model performance.

In summary, the main contributions of this paper are summarized as follows:

- We propose a reinforcement learning–based adaptive routing framework for mixture-of-experts (MoE) models, which dynamically predicts the number of activated experts per layer and token. This design enables flexible expert scheduling instead of relying on a fixed hyperparameter.

- We introduce a reward function that jointly considers accuracy, diversity, and efficiency. This unified formulation not only balances performance and computational cost but also alleviates the common problem of expert imbalance in MoE training.

- Extensive experiments on multiple LLM benchmarks demonstrate that our approach consistently reduces the number of activated experts and improves runtime efficiency, while maintaining or even slightly improving accuracy compared to strong MoE baselines.

## 2 THE APPROACH

We propose a reinforcement learning-based dynamic routing approach for adaptive expert scheduling within the standard MoE framework. Our model architecture is illustrated in Figure 1, where a

policy network is embedded within the base MoE module. The policy network predicts the number of experts to activate at each layer, guiding the router to select the top-$K$ experts for each token's computation. Additionally, we design a reward computation module that evaluates the model's final outputs, per-layer expert counts, and expert selection strategy to generate reward signals. Based on these reward signals, we construct a preference dataset and employ direct preference optimization (DPO) Rafailov et al. (2023) to jointly optimize the router and policy network. In the following texts, we firstly present the preliminaries of the MoE-based Transformer, next illustrate the integration of the MoE and policy network, then present the reward computation approach, and finally the RL optimization process of our approach.

## 2.1 PRELIMINARY

In a standard Transformer, each self-attention layer is followed by a fully connected feed-forward network. In the MoE Transformer, this feed-forward module is replaced by a module composed of a router and multiple experts. Assume the input hidden matrix at layer $l$ is $\mathbf{H}^{(l)} \in \mathbb{R}^{T \times d}$, where $T$ is the sequence length and $d$ is the hidden dimension. For the $t$-th token representation $\mathbf{h}_t^{(l)}$, the router computes gating probabilities using the parameter matrix $\mathbf{W}_r^{(l)} \in \mathbb{R}^{d \times E}$ by

$$\mathbf{p}_t^{(l)} = \mathrm{softmax}\big(\mathbf{h}_t^{(l)} \mathbf{W}_r^{(l)}\big) \tag{1}$$

where $\mathbf{p}_t^{(l)} \in \mathbb{R}^E$ denotes the probability distribution over the $E$ experts (the probability for the $e$-th expert is $p_{t,e}^{(l)}$). The router selects the top $k$ experts with the highest probabilities, forming the set $\mathcal{S}_t^{(l)}$, based on the hyperparameter $k$. The MoE module then aggregates the selected experts' outputs by a weighted sum to produce a new hidden representation $\tilde{\mathbf{h}}_t^{(l)}$ through

$$\tilde{\mathbf{h}}_t^{(l)} = \sum_{e \in \mathcal{S}_t^{(l)}} p_{t,e}^{(l)} \mathrm{Expert}_e^{(l)}\big(\mathbf{h}_t^{(l)}\big). \tag{2}$$

where $\mathrm{Expert}_e^{(l)}(\cdot)$ denotes the feed-forward mapping function of the $e$-th expert in layer $l$. After MoE computation, the resulting $\tilde{\mathbf{h}}_t^{(l)}$ undergoes a linear projection with residual connection and is passed to the next layer. Apart from replacing the feed-forward stage with the MoE module, the architecture remains the same as the standard Transformer, including LayerNorm and multi-head self-attention. At the last layer, the model projects the hidden representation to a prediction distribution to produce the predicted token $\widehat{y}_t$.

## 2.2 MoE WITH POLICY NETWORKS

In standard MoE, the number of experts activated at each layer is controlled by the hyperparameter $k$, preventing dynamic adjustment based on input. This fixed strategy leads to over-computation for simple inputs and insufficient resources for challenging inputs. To address this issue, we introduce a policy network to predict the number of experts to activate for each layer and token. Specifically, the policy network is a two-layer fully connected neural network taking the current token's hidden representation $\mathbf{h}_t^{(l)} \in \mathbb{R}^d$ as input. The network produces $\mathbf{u}_t^{(l)} \in \mathbb{R}^{K_{\max}+1}$, where the $i$-th component scores activating $(i-1)$ experts and $K_{\max}$ caps the maximum activation. This process is formulated as

$$\mathbf{u}_t^{(l)} = \mathbf{W}_2^{(l)} \sigma\big(\mathbf{W}_1^{(l)} \mathbf{h}_t^{(l)} + \mathbf{b}_1^{(l)}\big) + \mathbf{b}_2^{(l)} \tag{3}$$

where $\mathbf{W}_1^{(l)} \in \mathbb{R}^{h \times d}$ and $\mathbf{W}_2^{(l)} \in \mathbb{R}^{(E+1) \times h}$ are weight matrices, $\mathbf{b}_1^{(l)} \in \mathbb{R}^h$ and $\mathbf{b}_2^{(l)} \in \mathbb{R}^{E+1}$ are bias vectors, and $\sigma$ is the ReLU activation function. Subsequently, we apply softmax to convert $\mathbf{u}_t^{(l)}$ into a probability distribution, where the $i$-th value denotes the probability of activating $(i-1)$ experts. Afterwards, we predict the expert count $k_t^{(l)}$ accordingly, and the count determines the number of experts the router activates for the $t$-th token at layer $l$. Finally, our approach follows the standard computation process in MoE to aggregate the selected experts' outputs.

## 2.3 MODEL OPTIMIZATION WITH RL

To optimize the dynamic routing strategy, we employ an RL objective that improves prediction accuracy while dynamically adjusting the number of experts activated at each layer. Since the reward

function plays an essential role in RL, in the following texts, we firstly present the process to compute the reward and then illustrate the entire RL process.

**Reward Computation**     Given an input $\mathcal{X}$ with ground truth $\mathcal{Y}^*$, the LLM with our MoE produces a prediction $\widehat{\mathcal{Y}} = \widehat{y}_1 \cdots \widehat{y}_T$. We record the activated expert count $k_t^{(l)}$ and the selected expert set $\mathcal{S}_t^{(l)}$ for each token $t$ at each layer $l$. We define three reward functions to assess output accuracy, activation diversity, and computational efficiency, which are defined as $R_{\text{acc}}$, $R_{\text{div}}$, and $R_{\text{eff}}$, respectively. The details of the rewards are illustrated as follows.

The accuracy reward $R_{\text{acc}}$ measures consistency between the model prediction and the ground truth using cross-entropy, and is computed by

$$R_{\text{acc}} = -\text{Cross-Entropy}\big(\widehat{\mathcal{Y}}, \mathcal{Y}^*\big) \tag{4}$$

In general, a lower cross-entropy score indicates a better semantic alignment between the generated output and the ground truth.

The diversity reward is designed to prevent a few experts from shouldering all computations. Specifically, we compute the Kullback-Leibler (KL) divergence between the actual activation distribution $\mathbf{q}^{(l)}$ and the ideal uniform distribution $\mathbf{u}_e$ at each layer, then average these divergences across layers by

$$R_{\text{div}} = -\frac{1}{L} \sum_{l=1}^{L} \text{KL}\big(\mathbf{q}^{(l)} \,\|\, \mathbf{u}\big) \tag{5}$$

Herein, the average activation probability of expert $e$ at layer $l$ (denoted as $\mathbf{q}_e^{(l)}$, i.e., the probability for the expert $e$ in $\mathbf{q}^{(l)}$) is computed by

$$\mathbf{q}_e^{(l)} = \frac{1}{T} \sum_{t=1}^{T} \frac{\mathbb{I}\big[e \in \mathcal{S}_t^{(l)}\big]}{k_t^{(l)}} \tag{6}$$

where $\mathbb{I}[e \in \mathcal{S}_t^{(l)}]$ is an indicator function equal to 1 if expert $e$ is selected for token $t$ at layer $l$, and 0 otherwise.

The efficiency reward encourages the average activated expert count $\bar{k}$ to match the target $\kappa E$, where $E$ is the total number of experts, $\kappa \in (0, 1)$ is the desired activation ratio (e.g., $\kappa = 0.1$ means activating 10% of experts) and $\bar{k}$ is computed by

$$\bar{k} = \frac{1}{T \cdot L} \sum_{l=1}^{L} \sum_{t=1}^{T} k_t^{(l)} \tag{7}$$

Thus, the efficiency reward $R_{\text{eff}}$ is computed by

$$R_{\text{eff}} = \kappa E - \bar{k} \tag{8}$$

where the reward is positive when $\bar{k} < \kappa E$ and negative otherwise:

The overall reward $R$ is a weighted sum to balance all three objectives:

$$R = \lambda_{\text{acc}} R_{\text{acc}} + \lambda_{\text{div}} R_{\text{div}} + \lambda_{\text{eff}} R_{\text{eff}} \tag{9}$$

where $\lambda_{\text{acc}}, \lambda_{\text{div}}, \lambda_{\text{eff}} \geq 0$ are the hyperparameter weights for the accuracy, diversity, and efficiency rewards, respectively.

**RL Optimization**     For RL optimization, for each input $\mathcal{X}$, we firstly generate $N$ candidate outputs $\{\widehat{\mathcal{Y}}^{(n)}\}_{n=1}^{N}$, each associated with expert counts $\{k_t^{(l,n)}\}$ and expert sets $\{\mathcal{S}_t^{(l,n)}\}$. Then, we score all candidate outputs with a reward function and rank them accordingly. We mark the top-ranked candidate as the accept sample and the lowest-ranked as the reject sample. Based on accept–reject pairs, we build a preference dataset. Afterwards, we train the policy network and router using direct preference optimization (DPO) Rafailov et al. (2023) to maximize the relative probability of the accepted samples and minimize that of the rejected samples. This optimization process achieves end-to-end joint optimization of LLM output generation and expert routing.

Table 1: The statistics of the datasets used in pretraining and fine-tuning different models, where the number of examples and tokens are reported.

| Stage | Dataset | # Examples | # Tokens |
|---|---|---|---|
| Pre-training | Wikitext-103 | 1.8M | 101M |
| Fine-tuning | Stanford Alpaca | 52K | 3.0M |

Table 2: The summarization of our settings, where the model variants and the model initialization strategies are presented. The "✓" in Table (b) means that the parameters of the particular module are initialized by pre-trained LLMs, whereas "×" means the parameters are randomly initialized.

(a) Model variants

| Model | Architecture | Training Data |
|---|---|---|
| MoE-base | Standard MoE | Pre-training |
| Ours-PT | Ours | Pre-training |
| Ours–SFT | Ours | Fine-tuning |
| Ours-PT–SFT | Ours | Fine-tuning |

(b) Initialization strategies for the models

| Strategy | Transformer | MoE |
|---|---|---|
| Dense-partial | ✓ | × |
| Dense-full | ✓ | ✓ |
| MoE-full | ✓ | ✓ |

## 3 EXPERIMENT SETTINGS

### 3.1 DATASETS

To evaluate our approach in both pretraining and fine-tuning phases, we prepare three datasets spanning large-scale language modeling and instruction following. In the pretraining phase, we use Wikitext (Merity et al., 2016)[2] with high-quality Wikipedia articles. For the fine-tuning phase, we employ the Stanford Alpaca dataset (Wang et al., 2022)[3] with instruction-response pairs. Table 1 summarizes the statistics of the data used for the experiments. For evaluation, we use three widely adopted benchmarks, namely, MMLU (Hendrycks et al., 2021)[4], BIG-Bench (Srivastava et al., 2023)[5], and GSM8K (Cobbe et al., 2021)[6] to assess model performance on knowledge-intensive reasoning and problem-solving tasks. Specifically, MMLU is a multiple-choice benchmark covering 57 academic subjects across STEM, humanities, and social sciences. BIG-Bench is a large-scale collection of diverse and challenging tasks designed to probe broad capabilities of LLMs. We only use the multi-choice part of BIG-Bench to evaluate different models. GSM8K is a math word problem dataset that requires step-by-step reasoning to solve grade-school arithmetic problems. We use the multi-choice version of the GSM8K in experiments.

### 3.2 SETTINGS

In practice, large-scale MoE models are usually trained and applied in both pretraining and fine-tuning stages. Therefore, we evaluate our approach across these stages to assess whether it is able to consistently work under different conditions. Specifically, in the experiments, we compare four model variants that are illustrated as follows.

---

[2]https://huggingface.co/datasets/Salesforce/wikitext
[3]https://huggingface.co/datasets/tatsu-lab/alpaca
[4]https://huggingface.co/datasets/cais/mmlu
[5]https://huggingface.co/datasets/google/bigbench
[6]https://huggingface.co/datasets/guipenedo/gsm8k-mc

Table 3: The prompt used to instruct the models to perform multi-choice evaluation on different benchmark datasets. The placeholders for the question and the answer candidates are represented in brackets (in this example, there are four answer candidates). We use the same prompt for all models under different settings in the experiments.

*[Question]*
*A. [answer candidate A]*
*B. [answer candidate B]*
*C. [answer candidate C]*
*D. [answer candidate D]*
*Among A, B, C, and D, the answer to the question is*

- **MoE-base**: the model is obtained by pretraining a standard MoE-based LLM without using our approach on the pretraining corpora.

- **Ours-PT**: the model incorporates our adaptive routing approach into the MoE-base model and is pretrained on the same data.

- **Ours–SFT**: the model fine-tunes the MoE-base model on the fine-tuning dataset using our approach.

- **Ours-PT–SFT**: the model is obtained by first pretraining and then fine-tuning on the supervised data with our approach.

For the models, we examine three initialization strategies to analyze the impact of different model initialization approaches.

- **Dense-partial**: in this setting, the backbone model (i.e., the LLM) is initialized from a pretrained dense LLM, and the MoE modules remain randomly initialized.

- **Dense-full**: we initialize the backbone model from a pretrained dense LLM. In particular, for each expert, we copy the feed-forward weights from the corresponding fully connected layer of the dense LLM as the initial value. To prevent all experts from being identical, we inject independent small-magnitude Gaussian random noise and perturbations into each expert's weights. This preserves the initialization quality of the dense model while encouraging diversity across experts.

- **MoE-full**: in this setting, we use a pretrained MoE LLM as the backbone model.

The settings of different models and initialization strategies are summarized in Table 2.

### 3.3 IMPLEMENTATION DETAILS

In the main experiments, we run multiple LLMs to evaluate our approach under different initialization strategies. For the dense-partial and dense-full settings, we employ Qwen-3 0.6B (Yang et al., 2025)[7] and LLaMA-3.2 1B (Grattafiori et al., 2024)[8] pretrained dense LLMs as the backbone initializations. Specifically, under Qwen-3 0.6B, the model consists of 28 Transformer layers, a hidden dimension of 1,024. For LLaMA-3.2 1B, the model is configured with 16 Transformer layers, a hidden dimension of 2048. For models initialized with Qwen-3 0.6B or LLaMA-3.2 1B, we use 16 experts per layer, and a default activation count of $k = 4$. In addition, for the MoE-full initialization, we used the OLMoE-1B-7B-0125 (Muennighoff et al., 2024)[9] model as the MoE-base model. In this setting, we use the default architecture of the OLMoE-1B-7B-0125, which has 16 Transformer layers, a hidden dimension of 2048, 64 experts per layer, and a default activation count of 8. The hyper-parameter $\kappa$ to control the target number of the activation expert is set to 0.7, and the values of $\lambda_{acc}$, $\lambda_{div}$, and $\lambda_{eff}$ are all set to 1 for equal contributions of different rewards. In the pretraining phase, MoE-base and Ours-PT models are trained with a learning rate of 1e-4 and a batch size of 32. In the fine-tuning phase, Ours-SFT and Ours-PT-SFT models are trained with a learning rate of 2e-5 and a batch size of 16. For evaluation, we design a prompt illustrated in Table 3 to instruct

---

[7]https://huggingface.co/Qwen/Qwen3-0.6B

[8]https://huggingface.co/meta-llama/Llama-3.2-1B

[9]https://huggingface.co/allenai/OLMoE-1B-7B-0125

Table 4: Overall performance (i.e., accuracy) comparison of models with different settings (using different model variants, initialization strategies, and different pre-trained LLMs). "Avg." is the average performance on MMLU, BIG-Bench, and GSM8K. "Act. Ratio" indicates the ratio of average activated experts relative to the default setting.

| Model | Init Strategy | MMLU | BIG-Bench | GSM8K | Avg. | Act. Ratio |
|---|---|---|---|---|---|---|
| (a) Qwen-3 0.6B | | | | | | |
| MoE-base | Dense-partial | 39.1 | 30.0 | 29.6 | 34.6 | 1.0 |
| Ours-PT | Dense-partial | 39.7 | 29.5 | 29.9 | 34.6 | 0.7 |
| Ours-SFT | Dense-partial | 40.0 | 30.5 | 30.5 | 35.3 | 0.7 |
| Ours-PT-SFT | Dense-partial | 40.1 | 31.3 | 30.4 | 33.9 | 0.7 |
| MoE-base | Dense-full | 39.4 | 31.0 | 29.9 | 33.4 | 1.0 |
| Ours-PT | Dense-full | 39.6 | 31.3 | 29.7 | 33.5 | 0.7 |
| Ours-SFT | Dense-full | 40.1 | 31.7 | 30.7 | 34.2 | 0.7 |
| Ours-PT-SFT | Dense-full | 40.2 | 32.0 | 30.9 | 34.4 | 0.7 |
| (b) LLaMA-3.2 1B | | | | | | |
| MoE-base | Dense-partial | 36.4 | 27.2 | 28.4 | 30.7 | 1.0 |
| Ours-PT | Dense-partial | 36.6 | 27.3 | 28.5 | 30.8 | 0.7 |
| Ours-SFT | Dense-partial | 36.7 | 27.5 | 28.9 | 31.0 | 0.7 |
| Ours-PT-SFT | Dense-partial | 36.9 | 27.8 | 29.3 | 31.3 | 0.7 |
| MoE-base | Dense-full | 36.6 | 27.4 | 28.5 | 30.8 | 1.0 |
| Ours-PT | Dense-full | 36.8 | 27.5 | 28.7 | 31.0 | 0.7 |
| Ours-SFT | Dense-full | 37.0 | 27.7 | 28.7 | 31.1 | 0.7 |
| Ours-PT-SFT | Dense-full | 37.1 | 27.9 | 28.9 | 31.3 | 0.7 |
| (c) OLMoE-1B-7B-0125 | | | | | | |
| MoE-base | MoE-full | 40.2 | 32.3 | 30.2 | 34.2 | 1.0 |
| Ours-PT | MoE-full | 40.4 | 32.5 | 30.1 | 34.3 | 0.7 |
| Ours-SFT | MoE-full | 40.7 | 32.8 | 30.6 | 34.7 | 0.7 |
| Ours-PT-SFT | MoE-full | 41.0 | 33.0 | 30.8 | 34.9 | 0.7 |

the model to perform the multi-choice tasks on MMLU, BIG-Bench, and GSM8K. We compute the probabilities of predicting the answer candidates and regard the one with the highest probability as the model output. Following the evaluation convention of existing studies on the datasets, we report the accuracy of the models. We also report the ratio of the actual average activated expert count to the default activation count to quantify computational efficiency gains.

# 4 RESULTS AND ANALYSIS

## 4.1 OVERALL RESULTS

The performance of models with different initialization strategies and model variants is presented in Table 4, where different pre-trained LLMs are used. There are the following observations. First, compared to the MoE-base model, our approach under pretraining (PT), fine-tuning (SFT), and combined (PT-SFT) consistently achieves nearly the same or even better performance while substantially reducing the activation ratio. This indicates that our approach improves computational efficiency (in terms of the average number of activated experts) while maintaining model accuracy. Second, our approach remains consistently effective across different initialization strategies, including Dense-partial, Dense-full, and MoE-full. This demonstrates the robustness of our approach and its adaptability to models with different initialization qualities. Third, when comparing dense initialization strategies, we observe that Dense-full consistently outperforms Dense-partial. This trend is validated on models with Qwen-3.0 0.6B or LLaMA-3.2 1B backbones, showing that Dense-full initialization better leverages pretrained parameters. Fourth, under the MoE-full initialization, the model achieves overall stronger performance compared to other settings. This is because MoE-full directly uses a pretrained MoE model as the backbone, fully exploiting the benefits of sparse activation.

Table 5: Ablation study of different rewards, where OLMoE-1B-7B-0125 is used. "Full" refers to our full approach with all three rewards.

| | MMLU | BIG-Bench | GSM8K | Avg. | Act. Ratio |
|---|---|---|---|---|---|
| Full (Ours) | 41.0 | 33.0 | 30.8 | 34.9 | 0.7 |
| w/o $R_{acc}$ | 24.4 | 24.8 | 23.6 | 24.3 | 0.7 |
| w/o $R_{div}$ | 40.5 | 32.6 | 30.5 | 34.5 | 0.7 |
| w/o $R_{eff}$ | 40.7 | 32.8 | 30.6 | 34.7 | 1.0 |

## 4.2 ABLATION STUDY ON REWARDS

To investigate the contribution of different rewards to model performance, we conduct an ablation study. In each run, one reward component ($R_{acc}$, $R_{div}$, $R_{eff}$) is removed, and the results are reported in Table 5 based on the OLMoE backbone. We obtain several key observations. First, removing the accuracy reward $R_{acc}$ leads to a severe drop in prediction accuracy, with performance nearly reduced to random guessing. Second, removing the diversity reward $R_{div}$ causes highly imbalanced expert usage, where a few experts dominate the computation, resulting in a moderate performance drop compared to the full model. Finally, removing the efficiency reward $R_{eff}$ prevents the model from reducing the average number of activated experts, thereby losing its computational efficiency gains. These results demonstrate that the three rewards are complementary in routing optimization, jointly ensuring accuracy, balanced expert utilization, and efficiency.

## 4.3 VISUALIZATION

To better and more intuitively understand why our approach is effective, we conduct several visualizations in Figure 2. These include both the training process and the final expert activation distribution. For the training process, we visualize the loss curves and the number of activated experts over training steps in Figure 2(a) and (b), respectively. For the final model, we inspect the detailed activation distribution across all experts and layers in Figure 2(c). The details are illustrated as follows.

Figure 2(a) shows the training loss versus steps (in thousands) of MoE-base and Ours-PT-SFT. We observe that our approach (i.e., Ours-PT-SFT) converges faster than the baseline MoE-base, while the two models eventually reach similar loss values. This indicates that our adaptive routing improves optimization efficiency without sacrificing final accuracy. In practice, this faster convergence reflects that our approach reduces ineffective computation and emphasizes more informative gradients in early stages, which directly accelerates the optimization process.

Figure 2(b) illustrates the average number of activated experts in Our-PT-SFT over training steps. We observe that the number gradually decreases from the initial setting and eventually stabilizes near the target range (approximately five to six experts). This shows that the efficiency reward in our method successfully guides the model toward a compact activation budget. Importantly, the stabilization indicates that the model learns to maintain an effective balance: using fewer experts overall but still preserving accuracy and diversity of computation.

Figure 2(c) presents the heatmap of expert activations across all layers and experts in the final model (16 Transformer layers with 64 experts). The activations are distributed fairly evenly among experts, and no individual expert dominates or becomes overloaded. This demonstrates that the decrease in total activation is not achieved by repeatedly relying on a few specific experts. Instead, the model consistently selects relevant experts depending on the input, ensuring balanced utilization. Such balanced usage prevents overfitting of individual experts and improves robustness and generalization of the model.

## 5 RELATED WORK

MoE is a framework that distributes computational load across specialized subnetworks (i.e., "experts") via a gating mechanism (Shazeer et al., 2017). It achieves state-of-the-art performance on many downstream tasks and becomes the mainstream architecture choice for many LLMs (Artetxe et al., 2021; Shen et al., 2023; Yang et al., 2025; Meta, 2025). However, MoE models still face two

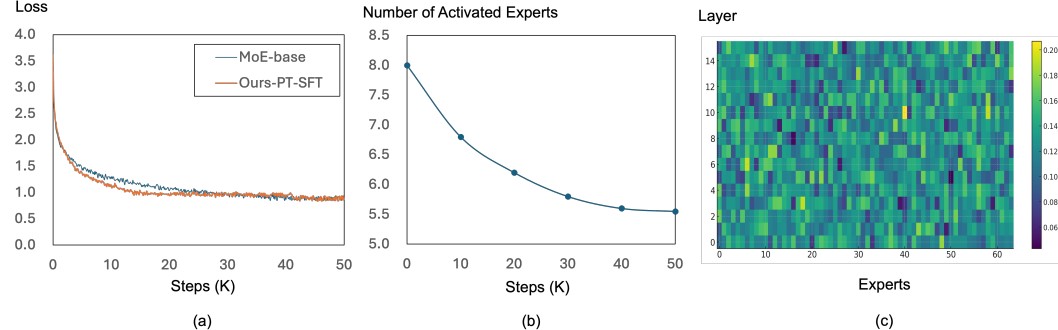

Figure 2: Visualization of our approach under the MoE-full initialization with OLMoE-1B-7B-0125 model. Figure (a) shows the training loss versus steps (in thousands), where our approach (Ours-PT-SFT) converges faster than the MoE-base while both models reach a similar final loss. Figure (b) illustrates the average activated experts vs steps (in thousands) under MoE-full initialization. Our approach (Ours-PT-SFT) shows a decreasing trend that stabilizes near the target, indicating effective control of activation budget. Figure (c) presents the expert activation heatmap across layers and experts (OLMoE-1B-7B-0125 has 16 Transformer layers with 64 experts), where the heatmap at each cell indicates the average chance of activating a particular expert at a particular layer.

key challenges: load imbalance (where a small subset of experts dominate activations) and static routing (which fixes the number of experts per token) (Zhou et al., 2022; Huang et al., 2024a; Wang et al., 2024; Zhang et al., 2025; Omi et al., 2025). To address the imbalance, Lepikhin et al. (2020) introduce an auxiliary loss to enforce uniform expert utilization, Fedus et al. (2022) propose expert capacity control to limit the number of tokens assigned to each expert. For dynamic expert selection, Raposo et al. (2024) extend MoE with a layer-skipping mechanism (Deep Mixture), allowing tokens to bypass layers deemed unnecessary based on complexity. Early approaches focus on content-aware gating: (Jin et al., 2024) use token difficulty estimates (derived from confidence scores) to adjust the count of expert activations, allocating more resources to harder inputs. Meanwhile, (Huang et al., 2024b) propose perceptual routing, where a lightweight classifier predicts token complexity and determines how many experts to activate. These approaches require separate training for difficulty estimation. Recent work directly integrates learnable routing strategies into the model. For example, (Jin et al., 2024) introduces "empty experts" as computational shortcuts; routers learn to select these placeholders for simple tokens, reducing the count of active experts. At the same time, (Zeng et al., 2024) proposes adaptive MoE (AdaMoE), where each layer's router dynamically adjusts expert activations based on the input's similarity to precomputed expert centroids. Although these approaches improve efficiency, they lack explicit mechanisms to balance performance and resource constraints, often requiring manual tuning of activation thresholds. For multi-objective optimization, Muzio et al. (2024) employ reinforcement learning (RL) to train routers to maximize accuracy while respecting latency budgets. Their approach treats expert selection as a sequential decision process, where rewards link to task performance and inference speed. Our work builds on these RL-based frameworks but introduces explicit diversity constraints and preference-based optimization to address expert imbalance and stabilize training. In the context of model efficiency, (Ong et al., 2024) uses preference data to train routers that balance accuracy and latency. Our approach differs by incorporating multi-faceted rewards (accuracy, diversity, efficiency) into preference signals, achieving joint optimization of task performance and resource allocation.

## 6 CONCLUSION

In this paper, we propose a reinforcement learning–based adaptive expert routing approach that improves the training and inference efficiency of MoE models while maintaining predictive accuracy. Specifically, we integrate a policy network into the standard MoE framework to predict the number of experts activated per token at each layer. We further design a multi-objective reward that balances accuracy, diversity, and efficiency, and optimize the system end-to-end using direct preference optimization. We validate our approach on multiple LLM benchmarks, showing that it achieves speedups and achieve even better performance, and demonstrates more balanced expert utilization and better applicability under resource-constrained settings.

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

## THEORETICAL FOUNDATIONS OF OUR APPROACH

We further analyze the theoretical foundations of the proposed approach, which are illustrated from the perspectives of optimal policy existence and information theory.

We begin with a global constrained optimization problem, whose objective is to balance prediction accuracy with computational cost. Let $\mathbf{k}_t = (k_t^{(1)}, \ldots, k_t^{(L)})$ denote the activation counts of experts for token $t$ across $L$ layers, and let $L_t(\mathbf{k}_t)$ denote the expected cross-entropy loss under this configuration. Therefore, the optimization problem is formulated as a combinatorial optimization with a global budget constraint:

$$\min_{\{\mathbf{k}_t\}_{t=1}^T} \sum_{t=1}^T L_t(\mathbf{k}_t) \quad \text{s.t.} \quad \frac{1}{TL} \sum_{l=1}^L \sum_{t=1}^T k_t^{(l)} \leq \kappa E \tag{10}$$

where $T$ is the number of tokens, $L$ is the number of layers, $E$ is the total number of experts, and $\kappa \in (0, 1)$ is the target activation ratio. This problem is essentially a combinatorial optimization with a global constraint, which is generally intractable to solve directly. A common approach to such problems is to apply Lagrangian relaxation, which converts the constraint into a penalty term. By introducing a multiplier $\lambda \geq 0$, the objective is reformulated as

$$\mathcal{L}(\{\mathbf{k}_t\}, \lambda) = \sum_{t=1}^T L_t(\mathbf{k}_t) \;+\; \lambda \left( \frac{1}{TL} \sum_{l=1}^L \sum_{t=1}^T k_t^{(l)} - \kappa E \right) TL \tag{11}$$

$$= \sum_{t=1}^T L_t(\mathbf{k}_t) \;+\; \lambda \sum_{l=1}^L \sum_{t=1}^T k_t^{(l)} \;-\; \lambda \kappa ETL \tag{12}$$

$$= \sum_{t=1}^T \left[ L_t(\mathbf{k}_t) + \lambda \sum_{l=1}^L k_t^{(l)} \right] \;-\; \lambda \kappa ETL. \tag{13}$$

In this relaxed form, the additional term $\lambda \sum_{l=1}^{L} k_t^{(l)}$ penalizes computation cost, while the "$-\lambda \kappa E T L$" term is a budget adjustment term. Thus, we obtain the dual problem:

$$g(\lambda) = \inf_{\{\mathbf{k}_t\}} \mathcal{L}\big(\{\mathbf{k}_t\}, \lambda\big) \tag{14}$$

$$= -\lambda \kappa E T L \ + \ \sum_{t=1}^{T} \inf_{\mathbf{k}_t} \left\{ L_t(\mathbf{k}_t) + \lambda \sum_{l=1}^{L} k_t^{(l)} \right\} \tag{15}$$

where $g(\lambda)$ denotes the dual function, defined as the optimal objective value of the relaxed problem under a given Lagrange multiplier $\lambda$. $\mathcal{L}(\{\mathbf{k}_t\}, \lambda)$ stands for the Lagrangian function, which consists of the original loss term and the penalty for computation budget. The operator $\inf$ refers to the infimum, meaning the minimal value over all possible activation configurations $\{\mathbf{k}_t\}$. We see that for fixed $\lambda$, the problem decomposes into independent token-level subproblems:

$$\mathbf{k}_t^* \in \arg \min_{\mathbf{k}_t \in \{0, \ldots, K_{\max}\}^L} \left\{ L_t(\mathbf{k}_t) + \lambda \sum_{l=1}^{L} k_t^{(l)} \right\} \tag{16}$$

where $\mathbf{k}_t^*$ represents the optimal activation configuration for token $t$, determined by balancing its prediction loss against the activation cost. This decomposition indicates that the global optimization is approximately reduced to local token-level optimization, which provides the theoretical basis for our approach. Furthermore, if we assume $L_t(\mathbf{k}_t)$ is approximately separable across layers, we obtain a layer-wise structure:

$$L_t(\mathbf{k}_t) \approx \sum_{l=1}^{L} L_t^{(l)}\big(k_t^{(l)}\big), \tag{17}$$

$$\mathbf{k}_t^* \in \arg \min_{\mathbf{k}_t} \sum_{l=1}^{L} \left\{ L_t^{(l)}\big(k_t^{(l)}\big) + \lambda \, k_t^{(l)} \right\}. \tag{18}$$

We then define the marginal improvement, which measures the loss reduction from activating one more expert

$$\Delta L_t^{(l)}(k) = L_t^{(l)}(k-1) - L_t^{(l)}(k). \tag{19}$$

In practice, $\Delta L_t^{(l)}(k)$ typically decreases as $k$ increases, reflecting diminishing returns. Therefore, the optimal allocation follows a threshold rule:

$$k_t^{(l),*} = \max \left\{ k : \Delta L_t^{(l)}(k) \geq \lambda \right\}. \tag{20}$$

This derivation shows that the policy network is viewed as a learnable approximation of the threshold rule, predicting the number of experts to activate given an input representation. In other words, our approach is essentially a specialization of the above optimization objective.

From an information-theoretic perspective, $k_t^{(l)}$ is interpreted as the "computation rate" allocated to token $t$ at layer $l$, while the loss $L_t^{(l)}(k)$ corresponds to a distortion measure. Thus, the global optimization problem is equivalent to an optimal rate allocation task under a budget constraint:

$$\min_{\{k_t^{(l)}\}} \frac{1}{T} \sum_{t=1}^{T} \frac{1}{L} \sum_{l=1}^{L} D_t^{(l)}\big(k_t^{(l)}\big) \quad \text{s.t.} \quad \frac{1}{TL} \sum_{l=1}^{L} \sum_{t=1}^{T} k_t^{(l)} \leq \kappa E \tag{21}$$

where $D_t^{(l)}(k)$ denotes the distortion measure, i.e., the prediction error when $k$ experts are activated for token $t$ at layer $l$. To solve this, we construct the Lagrangian:

$$\mathcal{L}(\{k_t^{(l)}\}, \lambda) = \frac{1}{TL} \sum_{l=1}^{L} \sum_{t=1}^{T} D_t^{(l)}\big(k_t^{(l)}\big) + \lambda \left( \frac{1}{TL} \sum_{l=1}^{L} \sum_{t=1}^{T} k_t^{(l)} - \kappa E \right) \tag{22}$$

$$= \frac{1}{TL} \sum_{l=1}^{L} \sum_{t=1}^{T} \left[ D_t^{(l)}\big(k_t^{(l)}\big) + \lambda k_t^{(l)} \right] - \lambda \kappa E \tag{23}$$

The notation $\mathcal{L}(\{k_t^{(l)}\}, \lambda)$ refers to the Lagrangian, which consists of all distortion terms plus the budget penalty. We observe that the optimization decomposes into independent subproblems at each token and layer:

$$k_t^{(l),*} \in \arg \min_{k \in \{0, \dots, K_{\max}\}} \left\{ D_t^{(l)}(k) + \lambda k \right\} \tag{24}$$

Here $k_t^{(l),*}$ denotes the optimal number of experts for token $t$ at layer $l$ under multiplier $\lambda$, i.e., the solution to the subproblem. Furthermore, if $D_t^{(l)}(k)$ is continuously differentiable, an approximate closed-form solution is obtained by the first-order condition:

$$\frac{\partial D_t^{(l)}(k)}{\partial k} + \lambda = 0 \quad \implies \quad k_t^{(l),*} \text{ is chosen such that the marginal distortion reduction equals } \lambda. \tag{25}$$

This corresponds exactly to the classical "water-filling" principle in information theory: the marginal reduction in distortion is equalized across all sub-channels. In this framework, the optimal allocation strategy is intuitively understood as assigning more computation resources to tokens with higher marginal gains and fewer to those with lower marginal gains. Moreover, our reward functions are also explained in this view: the accuracy reward $R_{\mathrm{acc}}$ corresponds to minimizing distortion, the diversity reward $R_{\mathrm{div}}$ corresponds to maximizing allocation entropy, and the efficiency reward $R_{\mathrm{eff}}$ corresponds to rate control under the global budget. Therefore, the combination of the policy network and reward functions is regarded as a learnable approximation of the water-filling principle, providing theoretical justification from an information-theoretic perspective.

From an information-theoretic perspective, $k_t^{(l)}$ is interpreted as the "computation rate" allocated to token $t$ at layer $l$, while the loss $L_t^{(l)}(k)$ corresponds to a distortion measure. Thus, the global optimization problem is equivalent to an optimal rate allocation task under a budget constraint:

$$\frac{1}{TL} \sum_{l=1}^{L} \sum_{t=1}^{T} k_t^{(l)} \le \kappa E \tag{26}$$

Within this framework, the optimal solution requires that the marginal reduction in distortion be balanced across tokens, which corresponds to the classical "water-filling" allocation rule (Cover, 1999), whose key idea is to achieve global optimality by equalizing marginal gains across different channels (or tasks). This principle indicates that under limited computational resources, the model should dynamically adjust the computation rate allocated to each token at each layer to ensure optimal resource utilization, which in turn provides theoretical justification for our approach. Furthermore, our reward functions also admit natural interpretations from this information-theoretic perspective: the accuracy reward $R_{\mathrm{acc}}$ corresponds to minimizing distortion, the diversity reward $R_{\mathrm{div}}$ is equivalent to maximizing the entropy of the activation distribution to avoid overload, and the efficiency reward $R_{\mathrm{eff}}$ reflects rate control under the global budget constraint. Therefore, in the information-theoretic sense, our approach is viewed as an instance of optimal rate allocation, where the policy network together with the designed rewards provides a learnable approximation to this principle, thereby ensuring the theoretical soundness of the approach.

