# OpenReview forum: "Reinforced Adaptive Routing for Mixture-of-expert Models"
_ICLR.cc/2026/Conference — ICLR 2026 Conference Withdrawn Submission_

### Official Review · Reviewer_7Q5c · 2025-10-27

**Soundness:** 1
**Presentation:** 2
**Contribution:** 1
**Rating:** 2
**Confidence:** 4

**Summary:**

This paper introduces a RL policy network into the training of MoE models, aiming to learn dynamic expert counts selected for each token at each layer. The authors claim that this approach improves efficiency, but I contend there are many factual mistakes.

**Strengths:**

The writing and figures are clear and well-organized.

**Weaknesses:**

1. The motivation lacks validity. Lines 13–15 state, “MoE requires activating multiple experts during training and inference, which introduces substantial computational and memory overhead.” However, it is important to note that MoE is explicitly designed to enable sparse activation compared to dense models with equivalent “total” parameter counts. The authors incorrectly frame MoE’s efficiency by comparing it to dense models with the same single MLP size. Deploying a sparsely activated model is typically more efficient. This suggests that the authors have a misunderstanding of MoE’s core design intent.

2. Based on this misunderstanding that MoE is inefficient, the authors propose a policy network to dynamically predict the number of activated experts per layer and per token, claiming improved efficiency due to lower activation ratios. However, this approach overlooks critical practical considerations. For instance, models are typically deployed on devices sized to handle their maximum computational and memory demands. If activation ratios vary dynamically across layers, layers with lower ratios will underutilize resources (whether memory- or compute-bound), leading to waste.

    The inefficiency issues of the proposed method might be exacerbated when activated expert counts are determined per token. In some cases, layer runtime is bottlenecked by the token requiring the most expert computations, while tokens assigned fewer experts finish early, creating pipeline bubbles. Additionally, deploying this approach with expert parallelism would introduce greater complexity in kernel optimization and communication engineering.

    It is recommended that the authors report training/inference runtime and throughput metrics. They should also explicitly clarify whether their implementation employs efficient MoE techniques—such as GEMM optimizations or expert parallelism—or relies on naive PyTorch for-loops. This distinction is critical: while naive loop-based implementations might easily yield apparent efficiency gains when reducing expert activation, real-world deployments leveraging GEMM and parallel computing often produce contradictory results.

3. The authors' claim of evaluating "pre-training" effectiveness using a mere 101M tokens is inconsistent with established practices in LLM pre-training and lacks validity. A 101M token dataset is not just "small" but orders of magnitude smaller than standard pre-training corpora. This "toy experiment" cannot mimic the long-term knowledge accumulation, generalization, and convergence dynamics of real pre-training.

4. The experimental improvements are insignificant. I do not think this method brings solid improvements in terms of downstream performance.

5. I might have missed it, but I did not find any discussion of the overhead introduced by the RL policy network. I do not know whether this additional module makes training slower than standard end-to-end training.

6. Figure 2a shows that between steps 15K and 40K, the training loss even increases slightly. Figure 2c shows that expert selection is not balanced at all: the most selected expert handles 20% of tokens, while the least selected processes only 6%. In practice, this leads to a pipeline bubble equivalent to the computation time for 14% of total tokens. How can this be considered efficient or balanced? Please provide quantitative metrics such as load balance loss [1] or balance entropy [2] to evaluate the model’s load balance.

[1] Eq.4, https://arxiv.org/pdf/2101.03961
[2] Eq.7, https://proceedings.mlr.press/v267/lv25b.html

**Questions:**

See weaknesses.

---

### Official Review · Reviewer_Z2eT · 2025-11-01

**Soundness:** 2
**Presentation:** 2
**Contribution:** 1
**Rating:** 2
**Confidence:** 5

**Summary:**

The paper proposes a reinforcement learning–based adaptive routing method for MoE)models. A policy network is introduced to dynamically predict the number of experts to activate per token at each layer. The routing strategy is trained via DPO, using a multi-objective reward that jointly considers accuracy, expert utilization balance, and computational efficiency.

**Strengths:**

The paper is clearly written and presents its method in a structured and understandable manner. The experimental setup includes multiple model backbones and benchmark tasks.

**Weaknesses:**

* **Significant Overlap with Prior Work**: The method proposed in this paper shows substantial similarity to a previously work at ICLR 2025 [1], particularly in terms of both architectural design and training strategy. In particular, the policy network used to predict the number of experts strongly resembles the allocator module in [1], and the formulation of the multi-objective reward also overlaps significantly. The main distinction appears to be the adoption of DPO in this work, as opposed to PPO in [1]. Given the considerable design overlap, I suggest the authors to explicitly acknowledge these similarities and clearly articulate how their method differs from, or improves upon, the prior work.

* **Evaluation Setup and Reported Performance**: The reported evaluation results raise questions regarding the interpretability of the proposed method. For instance, in the Table 4 (Row 5), when initializing from Qwen3-0.6B, the upcycled MoE trained with more data achieves 39.1 on MMLU and 29.6 on GSM8K, while the original Qwen3-0.6B [2] reports much higher scores (e.g., 52.81 on MMLU and 59.59 on GSM8K). The substantial performance gap casts doubt on the training effectiveness and model competitiveness. The authors are encouraged to provide more details about evaluation settings.

* **Lack of Empirical Evidence for Adaptive Routing**: The paper claims to support adaptive routing by dynamically adjusting the number of experts per token based on input complexity. However, the empirical evidence provided does not fully demonstrate this property. To substantiate the claim of adaptivity, the authors could provide (i) analyses of expert activation patterns across different tokens, (ii) performance breakdowns under different target sparsity levels (e.g., activation ratio κ).

[1] Yue T, Guo L, Cheng J, et al. Ada-k routing: Boosting the efficiency of moe-based llms[C]//The Thirteenth International Conference on Learning Representations. 2024.

[2] Yang A, Li A, Yang B, et al. Qwen3 technical report[J]. arXiv preprint arXiv:2505.09388, 2025.

**Questions:**

Please refer to Weaknesses.

---

### Official Review · Reviewer_jEjD · 2025-11-01

**Soundness:** 3
**Presentation:** 3
**Contribution:** 2
**Rating:** 2
**Confidence:** 4

**Summary:**

This paper introduces an adaptive routing method for MoE models. A policy network is introduced to allocate the number of experts for each token. It proposes a multi-objective reward function, optimized by DPO.

**Strengths:**

The proposed approach is conceptually intuitive and presented in a way that is accessible and easy to follow, making the methodology readily understandable.

**Weaknesses:**

1.The paper omits comparisons with some relevant baselines that share similar motivations. For example:
Mixture-of-Depth	[1], which achieves adaptive per-token computation by conditionally skipping layers.
AdaMoE[2], which enables dynamic expert selection through the use of null experts.
Including these baselines would strengthen the empirical analysis, especially given their focus on compute-adaptive architectures.

2.The paper does not report the distribution of expert activations across layers. If the computation is imbalanced layer-wise, pipeline parallelism becomes significantly more difficult to implement efficiently, affecting both training throughput and inference latency due to load imbalance and increased synchronization overhead.

3.Figure 2(c) is difficult to interpret and does not clearly demonstrate any discernible patterns. If the authors aim to support their claims regarding learned adaptive routing strategies, a more fine-grained and interpretable visualization is necessary.

[1] Raposo D, Ritter S, Richards B, et al. Mixture-of-depths: Dynamically allocating compute in transformer-based language models[J]. arXiv preprint arXiv:2404.02258, 2024.
[2] Zeng Z, Miao Y, Gao H, et al. Adamoe: Token-adaptive routing with null experts for mixture-of-experts language models[J]. arXiv preprint arXiv:2406.13233, 2024.

**Questions:**

Please refer to the above Weakness.

---

### Official Review · Reviewer_Ys1f · 2025-11-01

**Soundness:** 2
**Presentation:** 2
**Contribution:** 2
**Rating:** 2
**Confidence:** 4

**Summary:**

This paper introduces a reinforcement learning-based adaptive routing method for MoE models, aiming to improve computational efficiency while maintaining model performance. The approach incorporates a policy network to dynamically adjust the number of experts activated per token and layer, and optimizes the routing strategy using a multi-objective reward function via DPO.

**Strengths:**

The paper is well-structured and clearly written, with a coherent presentation of the motivation, method, and experimental design. The proposed method is evaluated across multiple model architectures and benchmark tasks, which strengthens the validity of the results.

**Weaknesses:**

1.Dynamic adjustment of the number of activated experts to enhance efficiency appears to have been recently and effectively explored by various works [1,2,3]. Given the high similarity in both the technical goal and the application scenario, this paper fails to clearly delineate its core adavantage and differentiation from this previous work through discussions or experiments.

2.Providing experiment data on the average latency/throughput could help make the inference cost improvement more significant. Besides, the paper does not demonstrate the relationship between the reduction in FLOPs and the activation ratio.

[1] Team M L C, Li B, Lei B, et al. Longcat-flash technical report[J]. arXiv preprint arXiv:2509.01322, 2025.

[2] Zeng Z, Miao Y, Gao H, et al. Adamoe: Token-adaptive routing with null experts for mixture-of-experts language models[J]. arXiv preprint arXiv:2406.13233, 2024.

[3] Yue T, Guo L, Cheng J, et al. Ada-k routing: Boosting the efficiency of moe-based llms[C]//The Thirteenth International Conference on Learning Representations. 2024.

**Questions:**

Please refer to Weakness.

---

### Note · Authors · 2025-11-13

I have read and agree with the venue's withdrawal policy on behalf of myself and my co-authors.